# CIRCUIT DISTILLATION

## ABSTRACT

Model distillation typically focuses on *behavioral* mimicry, where a student model is trained to replicate a teacher's output while treating its internal computations as a black box. In this work we propose an alternative approach: Distilling the underlying computational mechanisms implemented by a teacher model. Specifically, we propose *circuit distillation*, which introduces an objective to align internal representations between analogous circuit components in teacher and student models. We propose a method to match "functionally correspondent" circuit components and introduce a loss reflecting similarities between the representations that these induce. We evaluate circuit distillation on entity tracking and theory of mind (ToM) tasks using models from the `Llama3` family. Our results demonstrate that circuit distillation outperforms standard distillation, successfully transferring algorithmic capabilities by adjusting only a small, targeted subset of student model parameters. This work establishes the feasibility of transferring mechanisms, which may in turn allow for efficient distillation of targeted teacher capabilities via interpretable and controllable internal student mechanisms.

## 1 INTRODUCTION

*Model distillation* entails training a relatively small and efficient "student" LM using a larger and more capable teacher LLM (Gou et al., 2021). The prevailing training paradigm is one of behavioral mimicry: The student model is trained to replicate the output distribution of the large "teacher" model. This is typically done by minimizing the divergence between final-layer logits for the predictive task of interest (Hinton et al., 2015). More recent work has has focussed on distilling "reasoning" capabilities (Shridhar et al., 2023; Li et al., 2023; Wadhwa et al., 2024).

Distillation permits effective transfer of task-specific knowledge (Xu et al., 2024b). But the standard mechanism of knowledge transfer is fundamentally bottlenecked: The student learns only from teacher *outputs*, on the basis of which it must work out how to perform the task of interest. A more direct approach might be to allow the student to learn from the algorithms the teacher implements internally. This is the idea that we explore in this work.

More specifically, recent advances in *mechanistic interpretability* (Saphra & Wiegreffe, 2024; Sharkey et al., 2025) have established the existence of human-understandable algorithms implemented by subgraphs of attention heads and MLP layers within transformer models; these are called "circuits" (Shi et al., 2024). Instead of merely matching teacher outputs, we propose to guide the student model to functionally emulate the teacher's relevant circuit(s). Our hypothesis is that by enforcing this functional alignment at the component level, we can distill not just the teacher's knowledge, but its internal algorithms.

Following this intuition, we propose *circuit distillation*, in which we guide the student model to emulate a specific relevant internal mechanism to functionally align with the behavior of this circuit in a teacher model. This requires addressing two key technical challenges: (1) Establishing a functional correspondence between components in models of different sizes; (2) Designing an objective that enforces representational alignment between these circuits. We propose and evaluate approaches to these challenges and report promising empirical results. We offer the following contributions.

(1) We introduce *circuit distillation* (Figure 1), a new type of mechanistic distillation that modifies the student learning objective from output-level mimicry to direct alignment of internal circuits.

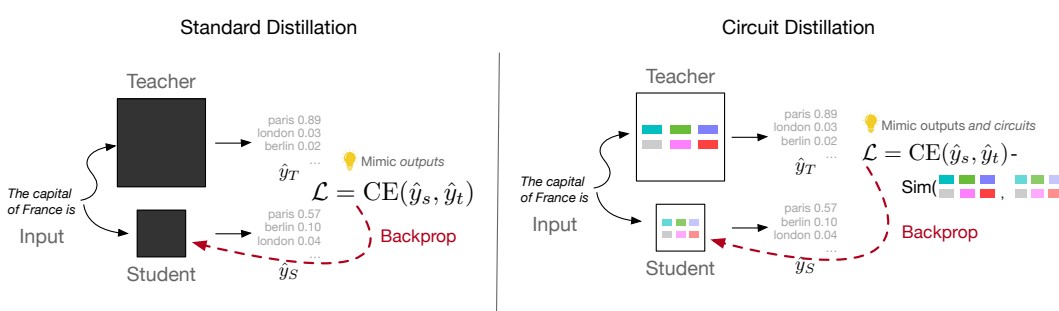

Figure 1: We propose *circuit distillation* which entails training a small student model to mimic not only the outputs of a larger teacher model (left; the standard distillation setting (Hinton et al., 2015)), but also its internal circuitry, i.e., *how* it executes a task of interest.

(2) We propose a functional component mapping strategy using *ablation impact similarity* to address the circuit correspondence problem, providing an intuitive method for identifying analogous attention heads between models of different scales.

(3) We introduce a transformation-invariant loss term to enforce representational similarity between the mapped circuit head representations during training.

(4) We validate this approach on two tasks: Entity tracking (Prakash et al., 2024) and causal Theory of Mind (ToM; Prakash et al. 2025). Our results show that mechanistic distillation can transfer mechanisms, enabling student models to perform tasks using internal mechanisms aligned with their teachers. This offers superior performance to models distilled using teacher outputs alone.

Our findings indicate that it is feasible to distill not just knowledge, but also the algorithms that produce it. By shifting the focus of distillation from outputs to circuits, we take a concrete step toward building models whose internal computations we can better understand and direct. This work may provide a foundation for future efforts investigating types of *mechanistic distillation*.

## 2 METHODS

In this section we describe our approach to model distillation from a mechanistic perspective. The goal is to induce the student model to emulate not only the teacher model's outputs, but also its internal mechanisms—and more specifically, relevant *circuits* (Conmy et al., 2023a; Shi et al., 2024). This requires first designing an approach to quantify representational similarity at the level of specific circuit heads (Section 2.1). Next we must integrate this similarity measure into a distillation objective term (Section 2.2) which we combine with the standard distillation loss (Section 2.3).

### 2.1 QUANTIFYING REPRESENTATIONAL SIMILARITY

Understanding and transferring learned algorithms from a teacher to a student model requires a method to compare their internal representations. In the context of circuits, we are particularly interested in the computations performed by specific, identifiable sub-components within the network. We focus on individual attention heads or MLP sub-layers that have been identified as participating in a circuit in the teacher model. Standard similarity metrics like cosine similarity between different (student and teacher) networks are not inherently meaningful given that dimensions may differ and are anyways arbitrary (e.g., due to basis rotations or isotropic scaling).

Therefore, we adopt Centered Kernel Alignment (CKA; Kornblith et al. 2019), which provides a robust and theoretically grounded measure of representational similarity between model internals. CKA is invariant to orthogonal transformations (including permutations) and isotropic scaling of the feature space, making it well-suited to compare activations from different architectures or—more relevant to our work—specific corresponding circuit heads in teacher and student of different sizes. This invariance allows us to compare the *functional similarity* of these circuit components.

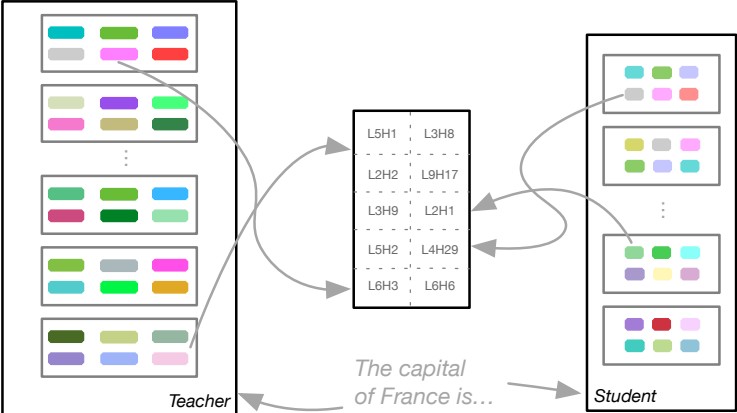

Figure 2: Circuit distillation requires mapping student heads to *functionally analogous* teacher heads. Here we propose do this by comparing performance drops under ablation (other approaches may also be viable). These correspondences are then used to align model components via the composite loss during distillation.

The intuition behind CKA is to measure the similarity between representational similarities induced by networks: Networks are "similar" if they produce representations that yield comparable pairwise similarities between examples (Kornblith et al., 2019). More precisely, given a batch of $m$ inputs, let $X \in \mathbb{R}^{m \times p_1}$ be the activation matrix from a specific circuit head in the student model, and $Y \in \mathbb{R}^{m \times p_2}$ be the activation matrix from the corresponding (or targeted) circuit head in the teacher model. Here $p_1$ and $p_2$ denote the dimensions of the activations for the respective circuit heads. We compute Gram matrices $K = XX^\top$ and $L = YY^{\top 1}$ and then measure correlations between the pairwise similarities induced by the two networks. Specifically we adopt the Hilbert-Schmidt Independence Criterion (HSIC), a kernel-based measure of independence between variables. Given two kernel matrices $K, L \in \mathbb{R}^{m \times m}$, and a centering matrix $H = I_m - \frac{1}{m}\mathbf{1}_m\mathbf{1}_m^\top$ (where $I_m$ is the $m \times m$ identity matrix and $\mathbf{1}_m$ is an $m \times m$ matrix of all ones), HSIC is calculated as:

$$\text{HSIC}(K, L) = \frac{1}{(m-1)^2}\text{tr}(KHLH) \tag{1}$$

The trace operator, $\text{tr}(\cdot)$, sums the diagonal elements of the resulting matrix. This adds the products of all corresponding entries in centered $K$ and $L$ matrices (which here are pairwise similarities between the respective network representations). Centering (via $H$) ensures that HSIC captures covariance structure, independent of means.

The reason not to use HSIC directly as our measure of representational similarity is that it is scale-dependent. This motivates CKA (Kornblith et al., 2019), which normalizes HSIC:

$$\text{CKA}(K, L) = \frac{\text{HSIC}(K, L)}{\sqrt{\text{HSIC}(K, K) \cdot \text{HSIC}(L, L)}} \tag{2}$$

CKA scores range from 0 (indicating that the representations are dissimilar, even after accounting for permissible transformations) to 1 (indicating that the representations are identical up to these transformations). This provides a scalar measure of how similarly two circuit heads process information across the input batch.

By focusing CKA on the activations produced by pre-identified circuit heads, we aim to measure the similarity of specific computational pathways rather than diffuse, layer-wide representational spaces. However, doing this requires a mapping of teacher circuit components to "comparable" student components; we next turn to how we do this.

---

[1] We use a linear kernel.

## 2.2 Aligning Corresponding Circuit Heads

We have a (differentiable) measure of similarity between activations, but which pairwise similarities should we enforce between student and teacher? What we want is to induce a circuit in the former akin to a relevant circuit in the latter. Operationally this requires aligning circuit components in the teacher to analogous components in the student, i.e., finding *functionally analogous* circuit heads. It is not obvious how to do this, particularly as student and teacher models will often differ in size (e.g., `Llama8B` teacher and a `Llama3B` student). We propose an ablation-based strategy to map student heads to teacher heads based on their respective contributions to task performance. This mapping then guides the CKA-based alignment.

The approach is as follows. First, we establish baseline performance metrics for both the student and teacher models on the target task dataset. Let $P_{\text{s\_base}}$ be the performance of the student model and $P_{\text{t\_base}}$ be that of the teacher model. We then quantify the functional importance of each relevant student head $h_s$ by recording the performance drop on its removal. We denote the student model's performance when head $h_s$ is ablated (i.e., its activations replaced with mean activations) by $P_{\text{s\_abl}}(h_s)$. The ablation impact for student head $h_s$ is then:

$$\Delta P_{\text{s}}(h_s) = P_{\text{s\_base}} - P_{\text{s\_abl}}(h_s) \tag{3}$$

Analogously, for each relevant teacher head ($h_t$), we record $P_{\text{t\_abl}}(h_t)$ or the teacher's performance when head $h_t$ is ablated. Crucially, to align with the student's operational capabilities, the teacher head's ablation impact is calculated *relative* to the student's baseline performance:

$$\Delta P_{\text{t}}(h_t) = P_{\text{s\_base}} - P_{\text{t\_abl}}(h_t) \tag{4}$$

This definition aims to identify teacher heads that, when ablated, result in performance degradation comparable in magnitude to that observed when ablating a student head, relative to student performance. With these ablation impacts computed, we create a mapping. For each student head $h_s$, we seek teacher heads $h_t$ whose ablation impact $\Delta P_{\text{t}}(h_t)$ is most similar to $\Delta P_{\text{s}}(h_s)$. For this we use the absolute difference:

$$d_{\text{abl}}(h_s, h_t) = |\Delta P_{\text{s}}(h_s) - \Delta P_{\text{t}}(h_t)| \tag{5}$$

Smaller $d_{\text{abl}}(h_s, h_t)$ indicates a greater similarity in functional importance as measured by ablation. We then map $h_s$ to the $h_t$ that minimizes $d_{\text{abl}}(h_s, h_t)$. This yields a mapping $\mathcal{M} : \mathcal{H}_{\text{s}} \to \mathcal{P}(\mathcal{H}_{\text{t}})$, where $\mathcal{H}_{\text{s}}$ and $\mathcal{H}_{\text{t}}$ are the sets of relevant student and teacher heads, respectively. This mapping $\mathcal{M}$ determines the pairs of student and teacher circuit heads that are used in the CKA loss term of the composite loss function (described below), thereby guiding the student model to align its internal mechanisms with those functionally important counterparts identified in the teacher.

## 2.3 A Composite Loss Function for Circuit Distillation

To align the student's circuit to the teacher's, we maximize the CKA between the corresponding heads. CKA values are 1 for perfect alignment (up to invariant transformations), so we define the CKA loss for a single pair of student ($K_{\text{s}}$) and teacher circuit head ($K_{\text{t}}$) Gram matrices as:

$$\mathcal{L}_{\text{CKA}}(K_{\text{s}}, K_{\text{t}}) = 1 - \text{CKA}(K_{\text{s}}, K_{\text{t}}) \tag{6}$$

Minimizing this loss pushes the CKA score toward 1, i.e., towards alignment of the induced representations under the corresponding circuit heads.

We then define a composite objective for the student model as a weighted combination of the primary task loss and the sum of CKA-based circuit similarity losses over aligned circuit head pairs:

$$\mathcal{L}_{\text{total}} = \mathcal{L}_{\text{task}}(y, \hat{y}_{\text{s}}) + \lambda \sum_{c \in \mathcal{C}_{\text{paired}}} \mathcal{L}_{\text{CKA}}(K_{\text{s}}^{(c)}, K_{\text{t}}^{(c)}) \tag{7}$$

Figure 3: Illustrations of the evaluated tasks: entity tracking (left), which tests recall of item locations, and theory of mind (right), which tests reasoning about beliefs under partial observability.

Where:

- $\mathcal{L}_{\text{task}}(y, \hat{y}_{\text{s}})$ is the conventional loss for the downstream task, with $y$ denoting the teacher labels and $\hat{y}_{\text{s}}$ the student predictions.

- $\mathcal{C}_{\text{paired}}$ denotes the set of pairs of circuit heads identified as analogous in the student and teacher models. The choice of these pairs is critical and should be informed by mechanistic understanding of which circuits in the teacher are most vital for the task or represent desirable computational properties.

- $K_{\text{s}}^{(c)}$ and $K_{\text{t}}^{(c)}$ are the Gram matrices derived from the activations of the $c$-th paired student and teacher circuit heads, respectively.

- $\lambda$ is a scalar hyperparameter that balances the contribution of the task performance objective against the mechanistic alignment objective. A careful tuning of $\lambda$ is necessary to ensure that the student model effectively learns the task while also internalizing the desired computational strategies from the teacher's circuits.

## 3 EXPERIMENTAL SETUP

Next we outline our experimental design, including the models, tasks, and the methods that we use to identify relevant circuit components for our mechanistic distillation study. The aim of these experiments is to assess whether and to what extent circuit distillation yields stronger performing student models, compared to standard distillation (based solely on teacher outputs). We also perform ablations to establish that the alignment between circuit components matters.

### 3.1 MODELS AND TASKS

We use models from the `Llama3` family (Dubey et al., 2024), specifically the 1B, 3B, and 8B parameter versions. The larger 8B model is used as the teacher and the smaller models as students.[2] This allows us to explore distillation across varying model capacities.

We focus on two tasks, depicted schematically in Figure 3. The selection of these models and tasks is deliberate: We are following recent mechanistic analyses of these tasks, which characterized relevant circuits in play (Prakash et al., 2024; Zhu et al., 2024). By using established circuits reported in prior work, we can focus on the challenge of transferring a known mechanism (circuit) from a teacher to a student. This avoids the separate and complex task of circuit discovery, which is an ongoing research direction in its own right.

The first task we consider is **Entity Tracking** (Figure 3, left). We adopt the dataset and setup from Prakash et al. (2024) to explore how models maintain and update information about entities within a given context. This task requires the model to identify the contents of a specific box based on a preceding context describing various objects in different boxes. For example, given "The keys are

---

[2]We use *relatively* small teacher and student models throughout this work owing to compute constraints.

in Box C, the phone is in Box A. Box C contains the...", the model should predict "keys". Motivated by the findings that models fine-tuned on arithmetic tasks (e.g., the GOAT corpus; Liu & Low 2023) show superior entity tracking capabilities, we first fine-tune the teacher model on GOAT.

The second task we consider is causal **Theory of Mind (ToM)** (Figure 3, right). This task investigates the model's ability to reason about characters' beliefs, especially when those beliefs depend on differing perspectives or access to information.

For this we create a dataset from prompts provided in Prakash et al. (2025). This dataset comprises simple stories where characters interact with objects, and the model must infer beliefs based on visibility and actions. For instance, a story might state: "Sarah places her book in the drawer and leaves the room. While Sarah is gone, Tom moves the book to the shelf." When subsequently prompted, "Where does Sarah believe her book is?", the model should infer Sarah's belief (the drawer) rather than the actual location (the shelf).

For ToM, prior work established that instruction-tuned `Llama3` models perform significantly better than their base counterparts.[3] The proprietary data used for official `Llama3`-Instruct models is not available, so we first prepare our teacher models by instruction-tuning larger `Llama3-8B` base variant on the publicly accessible Alpaca dataset (Taori et al., 2023). This custom instruction-tuned `Llama3` model then acts as the teacher.

## 3.2 IDENTIFICATION OF RELEVANT CIRCUIT HEADS

A key component of our proposed mechanistic distillation approach is identifying and targeting specific (teacher) circuit heads relevant to a given task. Methods for identifying such circuits are currently task-dependent, improving automated discovery of circuits is an active area of research (Conmy et al., 2023a; Wang et al., 2022). Here we rely on prior efforts that have characterized circuits for the two tasks we consider.

For entity tracking, we follow path patching as described in Prakash et al. (2024), which allows for the identification of a sparse set of attention heads that form the core circuit responsible for tracking entities and their properties. For the ToM task, where distinct belief-tracking and reasoning circuits are hypothesized, we use activation patching (with mean ablation) to assess causal impact of individual attention heads to identify the top-$n$ heads most critical for successful ToM reasoning. Details of these experiments and patching methods are provided in Appendix C. The sets of circuit heads identified through these procedures for both tasks subsequently serve as alignment targets in our CKA-based mechanistic distillation objective described in Section 2.

## 4 RESULTS

This section presents the empirical validation of our mechanistic distillation framework across the two tasks detailed in the previous section: Entity Tracking and ToM. Our analysis is structured to first establish baseline capabilities and the fidelity of the identified circuits within teacher and student models. We first identify relevant circuits in teacher models as proposed in prior work (Section 3.2) and then search for the corresponding circuits in the student models as described in Section 2.2. We then provide a direct comparison of standard behavioral distillation (considering both full model and targeted circuit updates; CE only) against our proposed mechanistic distillation method (CE + Align CKA) and a crucial control condition (CE + Rand CKA) to isolate the effect of principled circuit alignment. (We note that circuit distillation also affords efficiency gains: See Appendix Figure 5.)

### 4.1 ENTITY TRACKING

Table 1 summarizes the results on the entity tracking task, demonstrating the efficacy of aligning internal mechanisms. The "Full Model" column reports the accuracy achieved using the entire model, while "Circuit" measures performance when only the pre-identified entity tracking circuit is active, with all other attention heads mean-ablated.

We first establish performance benchmarks to contextualize our findings. The teacher model achieves an accuracy of 0.85 on this task. Importantly, the identified entity tracking circuit within the

---

[3]Our preliminary experiments corroborated these findings.

|  |  | Loss | Full Model | Circuit |
|---|---|---|---|---|
| Baselines | Base–1B | – | 0.68 | 0.65 |
|  | Base–3B | – | 0.72 | 0.70 |
|  | GOAT-8B | CE | **0.85** | **0.81** |
| Fully Distilled | Base–1B | CE | 0.75 | 0.66 |
|  | Base–3B | CE | 0.78 | 0.71 |
| Circuit Distilled | Base–1B | CE | 0.73 | 0.69 |
|  |  | Align CKA | 0.68 | 0.68 |
|  |  | CE + Rand CKA | 0.58 | 0.54 |
|  |  | CE + Align CKA | 0.77 | 0.73 |
|  | Base–3B | CE | 0.77 | 0.75 |
|  |  | Align CKA | 0.73 | 0.72 |
|  |  | CE + Rand CKA | 0.63 | 0.61 |
|  |  | CE + Align CKA | **0.82** | **0.79** |

Table 1: Entity tracking results for `Llama3` models. We report results for full distilation (all parameters) and circuit distillation (only matched circuit head parameters) under different losses: Cross Entropy (CE), Centered Kernel Alignment (CKA), CKA with Random Head Assignment, and CE + CKA with ablation-based head alignment loss. Chance accuracy is 0.14. The improvements of circuit distillation (1B, 3B) using CE + Align CKA are statistically significantly better than standard full distillation under CE and circuit distillation using CE only ($p<0.05$) under McNemar's test (details in Appendix).

teacher model operating in isolation nearly matches this, with an accuracy of 0.81. This high fidelity validates that the circuit discovered is indeed the primary locus of the entity tracking capability in the teacher model, making it a suitable target for distillation.

The base `Llama3-3B` student model begins with a lower baseline accuracy of 0.72; we aim to close the performance gap via distillation. Fine-tuning with traditional distillation improves its performance to 0.78 with full model updates and 0.77 with targeted circuit updates (which constitute approximately 11% of its total attention heads). We also observe that the latter yields a smaller gap between full model and circuit-only performance showing that our proposed circuit alignment method can identify relevant circuits within student models. The improvements in the students *circuit accuracy* suggest that standard CE loss can impart circuits to some degree, at least when we fine-tune only circuit relevant parameters.

Our proposed mechanistic distillation method `CE + Align CKA`, which uses a composite loss (Equation 6) on functionally aligned heads, achieves an accuracy of 0.82 (with 0.79 for the circuit), a substantial improvement compared to using the CE objective alone. Adding the CKA loss to explicitly mimic the teacher's circuitry nearly closes the gap between student and teacher when considering circuit-only performance.

We also assess the impact of explicit *alignment of circuit components* via CKA by randomly assigning each of the identified teacher heads to one of the student heads (`(CE + Rand CKA)`). This approach yields an accuracy of 0.63, harming performance relative to the CE baseline. This means the CKA loss—if applied to indiscriminate pairs—is not by itself helpful.

Taken together, these results suggest that enforcing representational alignment on a small, functionally critical subset of components facilitates a more direct and effective transfer of the underlying computational mechanisms compared to merely matching the teacher's final predictions.

## 4.2 THEORY OF MIND

To assess whether our framework generalizes beyond retrieval-based tasks to more complex reasoning problems, we apply the same methodology to the ToM task. The results, presented in Table 2, corroborate and extend our initial findings.

|  | | Loss | Full Model | Circuit |
|---|---|---|---|---|
| Baselines | Base–1B | – | 0.55 | 0.54 |
| | Base–3B | – | 0.58 | 0.56 |
| | 8B-Instruct (Meta) | – | **0.79** | **0.78** |
| | 8B-Alpaca | CE | 0.76 | 0.76 |
| Fully Distilled | Base–1B | CE | 0.60 | 0.55 |
| | Base–3B | CE | 0.63 | 0.56 |
| Circuit Distilled | Base–1B | CE | 0.59 | 0.55 |
| | | Align CKA | 0.52 | 0.47 |
| | | CE + Rand CKA | 0.49 | 0.44 |
| | | CE + Align CKA | 0.64 | 0.61 |
| | Base–3B | CE | 0.62 | 0.57 |
| | | Align CKA | 0.55 | 0.55 |
| | | CE + Rand CKA | 0.49 | 0.41 |
| | | CE + Align CKA | 0.65 | 0.65 |

Table 2: Theory of Mind (ToM) results for `Llama3` models. Instruction-tuned models (e.g., Llama3-Instruct) outperform their base counterparts. The improvements of circuit distilled (1B, 3B) using CE + Align CKA are statistically significantly better than standard full distillation under CE and circuit distillation using CE only ($p<0.05$) under McNemar's test (details in Appendix).

To establish a performance ceiling, we evaluate Meta's proprietary `Llama3-8B-Instruct` model, which achieves an accuracy of 0.79. Since the instruction-tuning data for this model is not publicly available, we trained our teacher model by fine-tuning a base Llama3-8B model on the Alpaca dataset. This `8B-Alpaca` model achieves an accuracy of 0.76. The base Llama3-3B model, our designated student, exhibits a limited innate capacity for causal ToM, achieving a baseline accuracy of 0.58 with its full architecture and 0.56 with its identified circuit components operating in isolation.

Our main experiments (Table 2; bottom row), in which we train only the identified ToM circuit heads, clearly illustrate the impact of each component of our approach. The behavioral distillation baseline using only a Cross-Entropy loss (`CE only`) improves the student's full model accuracy to 0.62 and its circuit accuracy to 0.57. In contrast, the control condition applying a CKA loss with randomly mapped heads (`CE + Rand CKA`) degrades performance to 0.49 (full model) and 0.41 (circuit), falling below the initial student baseline. This confirms that enforcing the alignment between functionally irrelevant components is detrimental to the distillation process. Finally, our mechanistic distillation method (`CE + Align CKA`) achieves an accuracy of 0.65 for both the full model and the isolated circuit. This demonstrates that the performance gains are successfully concentrated within the targeted circuit, confirming the efficacy of our functional alignment strategy.

## 5 RELATED WORK

Our work draws on two active research areas: Model distillation and mechanistic interpretability.

**Knowledge Distillation** was originally proposed as a model compression technique where a smaller student model is trained to mimic the soft-label output distributions of a larger teacher (Hinton et al., 2015). This paradigm has since evolved, with modern approaches often focusing on distilling complex capabilities into LLMs. A prominent strategy involves using powerful teacher models to generate large-scale instruction following datasets for fine-tuning smaller, open-source models (Taori et al., 2023; Chung et al., 2022).

Other work has moved beyond final answers to distill the reasoning process itself. By training students on Chain-of-Thought (CoT) explanations elicited from a teacher, researchers have improved student reasoning abilities (Wei et al., 2023; Li et al., 2024). Some methods have also explored matching intermediate representations (Park et al., 2024), though often at the level of entire layers rather than specific, functionally-defined components. Our work departs from these behavioral

and full-layer approaches by proposing a mechanistic view of distillation, where the objective is to directly transfer a known computational algorithm by aligning specific, pre-determined circuits.

**Mechanistic Interpretability** seeks to reverse-engineer neural networks into human-understandable algorithms (Olah et al., 2018). A key concept in this field is the "circuit," a subgraph of model components that implements a scrutable computation (Elhage et al., 2021; Wang et al., 2022). Researchers typically identify these circuits using causal tracing techniques, such as path patching or activation patching, to isolate the components that are causally responsible for a specific model behavior (Goldowsky-Dill et al., 2023; Conmy et al., 2023b). This approach has successfully uncovered circuits for a range of behaviors, from factual recall and editing (Meng et al., 2023) to more algorithmic tasks like indirect object identification (Wang et al., 2022). The success of these efforts suggests that many complex behaviors are not diffuse properties of the entire network but are instead implemented by localized and specialized subgraphs, making them viable targets for analysis and, as we propose, for targeted transfer.

**Circuits for Cognitive Phenomena.** The circuit hypothesis has proven particularly fruitful for investigating how LMs model complex cognitive phenomena like entity tracking and theory of mind. The ability of LMs to track entities, e.g., has been a focus of mechanistic inquiry (Feng & Steinhardt, 2024; Li et al., 2021). Our work directly builds on the findings of Prakash et al. (2024), who used path patching to identify a sparse, well-characterized circuit responsible for this capability in Llama-family models. Similarly, while many studies have evaluated Theory of Mind (ToM) from a behavioral perspective (Shapira et al., 2024; Kosinski, 2024; Le et al., 2019; Xu et al., 2024a; Strachan et al., 2024; Chan et al., 2024; Jin et al., 2024). Recent mechanistic work has begun to uncover the underlying computational patterns, such as the "lookback mechanism" for belief tracking (Prakash et al., 2025). These efforts have focussed on *discovery* and *analysis*. By relying on these previously discovered circuits, we focus our efforts on the *transfer* of a known mechanism. To our knowledge, this is the first work to propose mechanistic distillation, and to use distillation not simply for compression, but as a technique for targeted algorithmic transfer.

## 6 CONCLUSIONS

We have proposed an alternative to traditional model distillation: *Circuit distillation*, a mechanistic approach which aims to directly instill in a student model relevant internal algorithms or computations (*circuits*), as opposed to merely mimicking output behavior alone. Our approach to circuit distillation entails first identifying functionally similar corresponding circuit components in teacher and student models (using ablation-impact similarity metrics), and then adding a loss term based on representational similarity to guide the student to emulate the teacher's internal computations.

On two cognitive tasks—entity tracking and theory of mind—we showed that this technique outperforms standard behavioral distillation. Moreover, distilling circuit parameters *only* offers efficiency gains compared to standard distillation (in which all parameters are adjusted). Specifically, by targeting only a small fraction of the student model's components (11-15% of attention heads), we successfully instilled much of the teacher's task-specific capability, indicating that direct alignment of circuits is a more effective strategy for algorithmic transfer than output mimicry alone.

Our findings show that circuit distillation may provide a useful means of efficient model compression and a way to perform controlled distillation: By guiding a student model to adopt a known circuit, we open the avenue to building smaller models that are interpretable and controllable by design, insofar as they are trained to emulate particular pieces of teacher model functionality.

This work does have several limitations. Our method is contingent on the prior identification of a well-defined circuit in the teacher model; finding such circuits remains challenging and is an ongoing topic of research in mechanistic interpretability. Furthermore, our ablation-based approach to mapping components from teacher to student is heuristic; more sophisticated methods for identifying functional correspondence across architectures may yield further improvements. Our empirical analysis is also somewhat limited: We considered only a few tasks (for which well defined circuits have been identified), and used relatively small models owing to limitations in compute. Despite these limitations, we think this work points to interesting follow-up directions related to transferring targeted functionality and efficient model distillation that goes beyond output mimicry.

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

| | | Loss | Full Model | Circuit | Random Circuit | Faithfulness |
|---|---|---|---|---|---|---|
| Baselines | Base-1B | – | 0.68 | 0.65 | 0.03 | 0.96 |
| | Base-3B | – | 0.72 | 0.70 | 0.02 | 0.97 |
| | Base-8B | – | 0.73 | 0.72 | 0.01 | 0.97 |
| | GOAT-1B | CE | 0.77 | 0.69 | 0.03 | 0.89 |
| | GOAT-3B | CE | 0.83 | 0.75 | 0.04 | 0.90 |
| | GOAT-8B | CE | 0.85 | 0.81 | 0.02 | 0.89 |
| Fully Distilled | Base-1B | CE | 0.75 | 0.66 | 0.01 | 0.88 |
| | Base-3B | CE | 0.78 | 0.71 | 0.03 | 0.91 |
| | Base-1B (GOAT-style) | CE | 0.71 | 0.69 | 0.01 | 0.97 |
| | | Align CKA | 0.69 | 0.57 | 0.01 | 0.82 |
| | | CKA + Rand CKA | 0.62 | 0.55 | 0.02 | 0.88 |
| | | CE + Align CKA | 0.75 | 0.74 | 0.03 | 0.98 |
| | Base-1B | CE | 0.73 | 0.69 | 0.02 | 0.94 |
| | | Align CKA | 0.68 | 0.68 | 0.01 | 1.00 |
| | | CE + Rand CKA | 0.58 | 0.54 | 0.01 | 0.93 |
| | | CE + Align CKA | 0.77 | 0.73 | 0.01 | 0.94 |
| Circuit Distilled | Base-3B (GOAT-style) | CE | 0.76 | 0.71 | 0.02 | 0.95 |
| | | Align CKA | 0.72 | 0.71 | 0.02 | 0.95 |
| | | CKA + Rand CKA | 0.69 | 0.62 | 0.01 | 0.90 |
| | | CE + Align CKA | 0.77 | 0.77 | 0.01 | 1.00 |
| | Base-3B | CE | 0.77 | 0.75 | 0.02 | 0.98 |
| | | Align CKA | 0.73 | 0.72 | 0.00 | 0.99 |
| | | CKA + Rand CKA | 0.63 | 0.61 | 0.01 | 0.99 |
| | | CE + Align CKA | **0.82** | **0.79** | 0.00 | 0.98 |

Table 3: Extended results on the Entity Tracking task.

## A ADDITIONAL RESULTS

Tables 3 and 4 report additional comprehensive set of results, offering a granular analysis across different model scales and training paradigms. These tables include several key baselines and metrics designed to comprehensively evaluate our mechanistic distillation approach and validate the underlying circuits.

First, we establish a clear performance landscape. The `Base-1B/3B/8B` models represent the innate capabilities of the open-source `Llama3` family on these tasks. The fully fine-tuned versions on GOAT and Alpaca serve as a practical upper bound on entity tracking and theory of mind, respectively. They demonstrate the performance achievable with direct, supervised training on the target

|  | | Loss | Full Model | Circuit | Random Circuit | Faithfulness |
|---|---|---|---|---|---|---|
| Baselines | Base-1B | – | 0.55 | 0.54 | 0.02 | 0.98 |
|  | Base-3B | – | 0.58 | 0.56 | 0.01 | 0.97 |
|  | Base-8B | – | 0.71 | 0.68 | 0.02 | 0.95 |
|  | Base-1B-Instruct | – | 0.65 | 0.64 | 0.02 | 0.98 |
|  | Base-3B-Instruct | – | 0.69 | 0.67 | 0.01 | 0.97 |
|  | Base-8B-Instruct | – | 0.79 | 0.78 | 0.02 | 0.99 |
|  | Base-1B-Alpaca | CE | 0.58 | 0.53 | 0.03 | 0.91 |
|  | Base-3B-Alpaca | CE | 0.66 | 0.64 | 0.05 | 0.97 |
|  | Base-8B-Alpaca | CE | 0.76 | 0.0.71 | 0.02 | 0.93 |
| FullyDistilled | Base-1B | CE | 0.60 | 0.55 | 0.02 | 0.92 |
|  | Base-3B | CE | 0.63 | 0.56 | 0.01 | 0.89 |
| CircuitDistilled | Base 1B | CE | 0.59 | 0.55 | 0.01 | 0.93 |
|  |  | Align CKA | 0.52 | 0.47 | 0.01 | 0.90 |
|  |  | CKA + Rand CKA | 0.49 | 0.44 | 0.04 | 0.89 |
|  |  | CE + Align CKA | 0.64 | 0.61 | 0.03 | 0.95 |
|  | Base 3B | CE | 0.62 | 0.57 | 0.00 | 0.92 |
|  |  | Align CKA | 0.55 | 0.55 | 0.00 | 1.00 |
|  |  | CE + Rand CKA | 0.49 | 0.41 | 0.01 | 0.84 |
|  |  | CE + Align CKA | 0.65 | 0.64 | 0.01 | 0.98 |

Table 4: Extended results on the ToM task.

data, against which we can measure the efficacy of our distillation approaches. For example, the `GOAT-8B` model achieves an accuracy of 0.85 on entity tracking.

Before evaluating distillation, we validate the integrity of the circuits themselves using two crucial metrics. The **Faithfulness** score, calculated as the ratio of `Circuit Accuracy` to `Full Model Accuracy`, quantifies how much of a model's capability is captured by its identified circuit. Across all base and fine-tuned models, faithfulness remains high (0.89-0.97), confirming that the identified circuits are indeed the primary mechanisms responsible for underlying tasks. As a sanity check, the **Random Circuit** column reports the performance of a randomly selected set of model components of the same size. The near-zero accuracy (0.01-0.04) in all cases provides strong evidence that identified circuits are non-trivial, functionally cohesive units and not arbitrary collections of components.

The central comparison in our study is between traditional behavioral distillation and our mechanistic approach. The `Distilled-1B` and `Distilled-3B` models represent the former, where the entire student model is fine-tuned on the teacher's outputs using a standard Cross-Entropy (CE) loss. These models show solid improvement over their base counterparts (e.g. `Distilled-3B` reaching 0.78 accuracy on the entity tracking task), but the `Circuit Distilled` models consistently outperform this baseline.

## A.1   STATISTICAL SIGNIFICANCE TESTING

To verify that the observed performance improvements of our framework are statistically significant, we use McNemar's test. This non-parametric test is well-suited for comparing the predictions of two models on the same test set by analyzing their disagreements in a contingency table.

For each `Circuit Distilled` model (1B and 3B), we ran two separate significance tests:

1. We compared the predictions of our proposed method (`CE + Align CKA`) against the circuit-only behavioral baseline (`CE` in the `Circuit Distilled` block). This test assesses whether the addition of the CKA alignment term provides a significant benefit over standard training for the circuit.

2. We compared the predictions of `CE + Align CKA` against the full-model behavioral distillation baseline (`Distilled` with `CE` loss). This more stringent comparison evaluates whether training only the circuit mechanistically can significantly outperform training the entire model behaviorally.

In all comparisons across both the 1B and 3B student models, the resulting $p$-values were well below the standard significance threshold of $\alpha = 0.05$, ranging from 0.003 to 0.007, providing strong statistical evidence that the performance gains achieved by our mechanistic distillation framework are not attributable to random chance and represent a genuine improvement over the baseline methods.

## B   CIRCUIT IDENTIFICATION AND MODEL SELECTION

This section provides details on the methodology used to identify the entity tracking circuits that serve as the primary targets for our mechanistic distillation experiments. For this task, our approach is a direct adaptation of the Path Patching technique described by Prakash et al. (2024), which has been demonstrated to effectively discover sparse, causal circuits for entity tracking in Llama-family models. Path Patching is an iterative, causal-tracing method designed to identify the subgraph of model components—in this case, attention heads—responsible for a specific behavior. The core of the technique involves comparing two forward passes: A `clean run` on an original, unmodified task input, and a `corrupted run` on a counterfactual input where the information required to produce the correct answer has been removed or altered. For the entity tracking task, a corrupted input is generated by randomizing the query box, the object names, and their corresponding box labels, thus ensuring that the model cannot solve the task through simple heuristics.

The process aims to find paths in the computational graph that, when "patched" with activations from the corrupted run, cause the most significant degradation in the clean run's performance. A "patch" involves replacing the activation of a specific node (e.g., an attention head at a given token position) in the clean run with the corresponding activation from the corrupted run. The causal

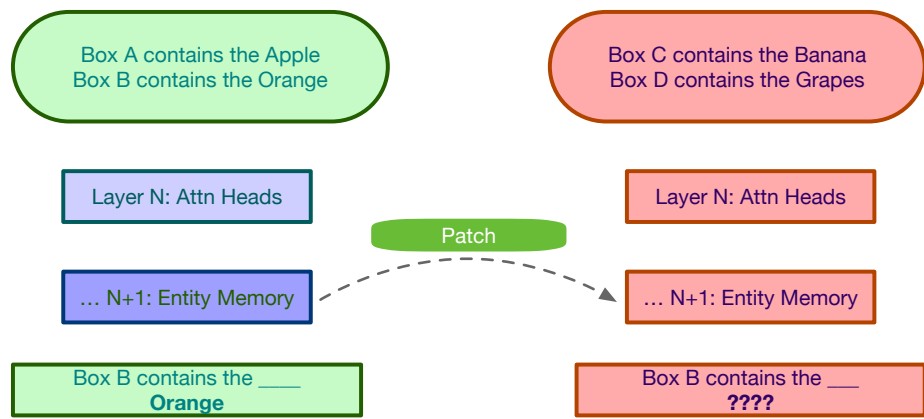

Figure 4: Path patching isolates which model components store and retrieve entity-attribute bindings.

importance of each path is quantified by a patching score, which measures the relative change in the log-probability of the correct output token when that path is patched. Paths yielding the largest performance drop (i.e., the most negative scores) are considered the most causally significant and are iteratively added to the identified circuit. This iterative search begins at the final token position to identify heads that directly influence the output logit (referred to as "Value Fetcher" heads in Prakash et al. (2024)). The search then proceeds backward through the model, identifying upstream heads that provide crucial information to the already-identified circuit components via query-key or value-vector compositions. This process reveals a sparse, multi-component circuit responsible for locating the query, transmitting positional information, and ultimately retrieving the correct entity value.

**Model selection**   Our decision to use GOAT-finetuned models as teachers for the entity tracking task is directly informed by the findings that emerge from this circuit analysis. Prakash et al. (2024) demonstrated that while a base `Llama` model possesses a nascent entity tracking circuit, fine-tuning on structured data like the arithmetic expressions in the GOAT dataset significantly enhances the functionality of this *same* underlying circuit. Specifically, the `Value Fetcher` and `Position Transmitter` components of the circuit become more precise and effective at resolving positional information and retrieving the correct object value. This finding provides a strong mechanistic basis for our experimental design: by selecting GOAT-finetuned models as teachers, we are targeting a known, more potent version of the very mechanism we aim to instill in the student model through distillation.

## C   EXPERIMENTAL DETAILS

All experiments were conducted on a single NVIDIA A100 GPU. To maintain experimental control and isolate the effects of the different loss functions, we used the default hyperparameters of the base `Llama3` models for all training runs. The only modification was a consistent learning rate of 2e-5, which was applied across all teacher fine-tuning, standard distillation, and mechanistic distillation experiments.

The datasets for our two tasks were generated as follows:

**Entity Tracking:**   The dataset was created by Kim & Schuster (2023), where they designed it to evaluate a model's ability to track the state changes of discourse entities. The dataset contains English sentences describing a setting where various objects are located in different boxes, and the task is to predict the contents of a queried box.

Following Prakash et al. (2024), we modify the structure of the context segment. Instead of the original format "Box F contains the apple," we reorder the phrases to "The apple is in box F." This structural change is crucial as it prevents the model from simply relying on shallow pattern matching

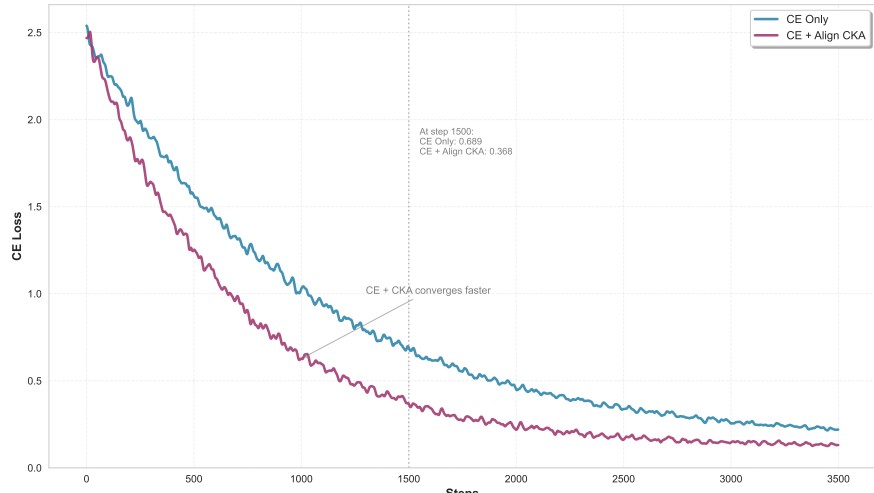

Figure 5: We compare the Cross-Entropy (CE) loss component for the `CircuitDistilled-3B` model on the entity tracking task when trained with a standard behavioral objective (CE Only) versus our mechanistic distillation method (CE + Align CKA)
.

or locating the longest identical context segment between the context and the query. It forces the model to genuinely infer the relationship between an object and its container. Each task instance in our setup involves several boxes, each labeled with a random letter and containing a unique, single-token object.

**Theory of Mind (ToM)**  For the ToM task, we use the CausalToM dataset (Prakash et al., 2025). This dataset was specifically constructed for the causal analysis of ToM reasoning, addressing the limitations of existing datasets which often lack the structure needed for counterfactual interventions. Each story involves two characters who each interact with a distinct object, causing it to take on a unique state (e.g., "Carla grabs an opaque cup and fills it with coffee. Then Bob grabs another opaque bottle and fills it with beer."). The model is then asked to reason about one character's belief regarding an object's state, often under conditions of limited or no visibility of the other character's actions.

## C.1 TRAINING DYNAMICS AND CONVERGENCE

An analysis of the training dynamics reveals that our mechanistic distillation approach not only achieves higher final accuracy but also learns more efficiently.

Figure 5 plots the cross entropy (CE) loss component during the training of the `CircuitDistilled-3B` model for entity tracking task, comparing the behavioral baseline (`CE Only`) against our mechanistic method (`CE + Align CKA`). The plot illustrates two key findings. First, the model trained with the `CE + Align CKA` objective converges faster. The composite loss provides a richer, more structured gradient signal that the CE loss alone, which accelerates the learning process. For example, at the 1500-step mark the CE loss for mechanistic run was nearly half that of the behavioral baseline. Second, the mechanistic approach also leads to a lower final CE loss value by the end of the training run.

We hypothesize that this improved efficiency stems from the nature of the CKA loss term. While the CE loss provides a sparse signal based only on the final output token, the CKA loss offers a denser gradient that guides the internal representations of the student's circuit to align with the teacher's. This forces the student model to not just find the correct answer, but to adopt a proven, effective internal algorithm for doing so, thereby regularizing the learning process and leading to a more efficient and effective convergence.

