# OpenReview forum: "Circuit Distillation"
_ICLR.cc/2026/Conference — Submitted to ICLR 2026_

### Official Review · Reviewer_8ybk · 2025-10-31

**Soundness:** 2
**Presentation:** 3
**Contribution:** 2
**Rating:** 4
**Confidence:** 4

**Summary:**

The paper proposes Circuit Distillation, a novel approach to model distillation that transfers mechanisms rather than behaviors. Instead of aligning full-model outputs, the method aligns specific task-related circuits between teacher and student networks. Using CKA-based similarity and ablation-derived functional mapping, the paper demonstrates that aligning these circuit heads improves transfer efficiency on two tasks: entity tracking and theory of mind.

**Strengths:**

- The idea of transferring mechanisms, rather than just outputs, is innovative yet simple

- The ablation-based mapping of student and teacher heads is simple

- Demonstrations on two nontrivial tasks (entity tracking and ToM) using Llama models go beyond toy examples and toy models; the ToM case is especially challenging.

- The approach makes a clear connection between interpretability research and practical distillation, hinting at possible efficiency and control benefits.

**Weaknesses:**

- Maun concern: The method distills task-specific circuits, not general capabilities. This limits its practical impact compared to standard knowledge distillation, which aims to transfer the global behavior of the teacher model.

- Since only single-task circuits are transferred, it’s unclear how multiple mechanisms would interact or accumulate—this is the most interesting setting.

- Initialization assumption: The author assumes the student already has a comparable circuit structure and is not randomly initialized, which is sometimes not the case in distilation.

- In all the experiments, the teacher–student gap is relatively small, making it hard to evaluate the results. Potentially, as a result, the gap between full distillation and circuit distilation is very small and not very impressive. unclear if the initial larger gap will make a better separation, or if both methods will still perform similarly.

- The paper does not test how distilling one circuit interferes with other capabilities of the student.

- Ablation-based mapping may not guarantee true functional correspondence. Heads with different functionalities can still have a similar effect on the performance degradation of the output when ablated.

**Questions:**

- Can multiple circuits be distilled jointly, or would they interfere?

- How does circuit distillation affect performance on unrelated tasks?

- Could this framework be applied in settings where the student is trained from scratch, without pre-existing circuits?

- Section 2.2 leaves ambiguity about whether the “performance” metric refers to final accuracy or the probability vector; clearer definitions would help.

- Minor comments:

Figure 2 caption: missing “to” in the phrase “to do” (second line).

---

> ### Author Response · Authors · 2025-11-13
> **Response to Reviewer #4**
>
> We thank the reviewer for their feedback and are glad they found the core idea innovative and simple, and our experiments nontrivial.
>
> The reviewer raises good high-level questions about the scope, limitations, and future scalability of our method, which we address below.
>
> **1. On Scope (W1, W2, W5) and Related Questions (Q1, Q2)**
>
> - **W1** (Task-specific, not general): The reviewer is correct that our work focuses on task-specific circuits. This was a deliberate choice. We note that there are many practical scenarios where one might want to distill functionality for a single specific downstream task, and we focussed on this sort of scenario to provide a first proof-of-concept that targeted mechanistic transfer is feasible and effective. We will add discussion regarding this to the manuscript.
>
> - **Q1 and Q2** (Multiple circuits / interference): We did not test for interference with unrelated tasks or the joint distillation of multiple circuits. We view these as interesting next steps for this line of research, and we agree with the reviewer that this may be "the most interesting setting." That said, this effort is intended to lay the groundwork for more complex (multiple) circuit distillation approaches. We will add discussion of this as a compelling direction for future work.
>
> **2. On Initialization (W3) and Training from Scratch (Q3)**
>
> - **W3 & Q3** (Student initialization): We agree our method is not suited for a randomly initialized student. We use standard pre-trained base models. We think that in practice, one would typically distill into a smaller pre-trained model; distilling into a randomly initialized model (from scratch) seems inefficient. So we think this is a reasonable assumption which would align with practice, and indeed our method requires the student to have some non-trivial baseline performance (shown in Tables 1&2) to make the mapping meaningful. We will explicitly state this assumption in Section 2.2.
>
> **3. On Performance Gaps (W4)**
>
> The reviewer notes the gap between full distillation and circuit distillation is "not very impressive." We believe this is a misunderstanding.
>
> Our key finding here is that we statistically significantly exceed the performance of full-model distillation (e.g., 0.82 vs 0.78 in Table 1, 4 absolute points performance gain) by fine-tuning only a small, targeted subset of parameters (11-15% of heads). This demonstrates a more efficient and effective transfer of the specific capability, which we believe is a strong result. We will re-emphasize this aspect in our results section.
>
> **4. On Methodological Details (W6, Q4)**
>
> - **W6** (Mapping fidelity): We agree our ablation-mapping is a heuristic and may not find the one, “true” or ideal functional map. However, our control experiment (CE + Rand CKA) confirms this heuristic is effective: our ablation-based map (CE + Align CKA) dramatically outperforms a random map (tab 1), proving its utility. We view the mapping strategy used here as just one option for the necessary step of alignment to realize circuit distillation, which is our main contribution.
>
> - **Q4** (Ambiguity in Section 2.2): Thank you for catching this. The 'performance' metric used for the ablation-impact mapping is the final task accuracy. We will explicitly define this in the text.
>
> We will also fix the typo in Figure 2 in the revised manuscript.

---

### Official Review · Reviewer_Mq9k · 2025-11-01

**Soundness:** 2
**Presentation:** 2
**Contribution:** 2
**Rating:** 4
**Confidence:** 2

**Summary:**

This paper introduces Circuit Distillation, a method that aims to transfer not only the output behavior of a large language model but also its internal reasoning mechanisms to a smaller student model. The approach first identifies functional “circuits” in the teacher model (i.e., specific groups of attention heads responsible for tasks like entity tracking or theory-of-mind reasoning), and then finds corresponding components in the student through ablation impact similarity and CKA-based representation alignment. During training, the student jointly optimizes standard cross-entropy loss and a circuit-alignment loss, encouraging it to reproduce the teacher’s internal computational patterns. Experiments on reasoning benchmarks show that Circuit Distillation outperforms traditional model distillation.

**Strengths:**

1. The paper proposes circuit distillation, which goes beyond traditional behavioral distillation by aligning internal circuit-level computations between teacher and student models
2. The formulation of the method (ablation-based mapping + CKA-based alignment) is well-defined
3. Results on entity tracking and theory of mind convincingly demonstrate that aligning internal circuits yields better transfer of algorithmic capabilities than standard distillation

**Weaknesses:**

1. The paper is quite difficult to follow. Key concepts such as “circuit mapping”, “ablation impact similarity”, and “CKA alignment” are introduced with minimal intuition or motivating examples
2. The ablation-based mapping between teacher and student heads is heuristic and potentially unstable
3. Computing ablation impacts for all heads across teacher and student is computationally heavy, and the paper does not report cost, runtime, or scalability to larger architectures
4. Experiments are confined to small-scale Llama 1B–8B models and narrow cognitive tasks

**Questions:**

1. What is the cost of computing ablations in circuit distillation?
2. Could the method scale to bigger sizes and different architectures?

---

> ### Author Response · Authors · 2025-11-13
> **Response to Reviewer #3**
>
> We thank the reviewer for their feedback, and are glad that they found the core idea of circuit distillation novel, techniques well defined, and the results convincing.
>
> The reviewer’s primary concerns relate to presentation (i.e. clarity) and practicalities of our mapping method (cost, scale, stability). We agree that the clarity of presentation can be improved in some places and will do so in revision, specifically by providing more intuition and motivation for the key concepts, as the reviewer suggests.
>
> We respond here in greater detail to the weaknesses raised by the reviewer below for clarity (but again will edit the manuscript to make these points clear in the paper).
>
> **Re key concepts in Sections 2.2 and 2.3:** The ablation impact similarity maps heads based on similar functional importance. For example, if ablating a teacher head causes a X% performance drop, we search for a student head whose ablation causes a similar drop and pair them. The CKA loss (Eq. 6) is the tool used to enforce this mapping. It trains the student's mapped head to align its representations with the teacher's.
>
> **Re Questions about mapping method (cost, scale, stability):** The reviewer raises good practical questions, and we agree that including these details in the paper would strengthen the work. We will add a new subsection (or Appendix) to address the omitted "Cost and Scalability" details, which are summarized as follows:
>
> 1. **Cost:** The mapping itself is a one-time, pre-compute step. For our Llama3-8B/3B models, this step took ~4 GPU-hours on one A100 with 400 instances from the entity tracking task. This is a fixed, one-off cost. We will add these runtime details to the paper.
>
>
> 2. **Stability:** Our mapping is a heuristic, but its effectiveness is what our experiments
> test.The results in Tables 1 and 2 (our CE + Align CKA vs. the CE + Rand CKA control) show this heuristic finds a meaningful correspondence insofar that it is far superior to random. We do not claim this is the possible mapping algorithm, of course; this step is necessary for realizing circuit distillation (our main contribution) but we are not especially committed to any particular mapping approach.
>
> 3. **Scale:** This is indeed a limitation. Our choice of relatively small models on tasks with known circuits (Sec 3.2) was deliberate for a controlled proof-of-concept. The mapping cost also scales with model/circuit size, but could be managed for larger models by using a smaller probing dataset or searching within a subset of layers. Our method is, in principle, applicable to other architectures and tasks, but here we have started with smaller models and tasks with known relevant circuits to as a first demonstration of the general technique.

---

### Official Review · Reviewer_QV1G · 2025-11-06

**Soundness:** 3
**Presentation:** 3
**Contribution:** 3
**Rating:** 6
**Confidence:** 2

**Summary:**

The authors propose to improve distillation of neural models by attempting to align the internal mechanisms (“circuits”) of the student and teacher models, rather than merely aligning outputs without the benefit of information about internal mechanisms. Specifically, the authors propose methods for “circuit distillation” which aim to change the student learning objective from output-level mimicry to direct alignment of internal circuits.

**Strengths:**

The intuition behind circuit distillation is sensible and compelling.

The authors examine the performance of various system components (e.g., CE, Align CKA, Random CKA) alone and in combination, to identify what components are necessary for that performance.

**Weaknesses:**

The introduction is somewhat difficult to follow, and does not immediately make a strong case for model distillation. The paper would be improved by a clearer and more compelling case for when model distillation is useful.

The testing is limited to only Entity Tracking and ToM. This is an extremely limited form of testing. The authors know this, and explain why, but it is still overly very limited evidence for effectiveness of the proposed methods.

As the authors note, the method assumes “the prior identification of a well-defined circuit in the teacher model” and “finding such circuits remains challenging and is an ongoing topic of research in mechanistic interpretability.” However, the basic approach outlined in the paper will only become more useful as (presumably) better methods for identifying circuits are developed.

**Questions:**

(none)

---

> ### Author Response · Authors · 2025-11-13
> **Response to Reviewer #2**
>
> We thank the reviewer for their positive assessment and for finding our approach compelling. We appreciate the constructive feedback on the paper's framing and limitations and will incorporate the following in the revised manuscript.
>
> 1. **On Introduction** – We will add a clearer, more compelling case for distillation itself grounding it in its primary goals of model compression, computational efficiency, and deploying capable models in resource-constrained environments, to beter motivate our mechanistic approach.
>
> 2. **Limited Testing** – We agree with the reviewer that our testing on only two tasks is a limitation. (As the reviewer notes, in the manuscript we do acknowledge this limitation and try to explain why these particular tasks make sense.) We do note that we find statistically significant improvements in performance on both tasks, and we think this does provide meaningful evidence for this new distillation paradigm’s utility. But this work is ultimately  a proof-of-concept which uses tasks where circuits were known (as noted in Section 3), allowing us to isolate and validate the transfer method itself, which we believe is a necessary first step.
>
> 3. **ON Dependency on Circuit Discovery** – We fully agree with the reviewer's assessment. Our method is indeed contingent on prior circuit identification, which is its main constraint (as we state in Section 6). We will reiterate this point in our conclusion, adopting reviewer’s framing that this work provides a foundational method for transferring circuit functionality, and its utility will naturally grow as discovery methods continue to improve.
>
>
> We thank the reviewer again for their thoughtful comments.

---

### Official Review · Reviewer_Dpyn · 2025-11-08

**Soundness:** 3
**Presentation:** 3
**Contribution:** 1
**Rating:** 2
**Confidence:** 4

**Summary:**

This paper proposes "circuit distillation" as an extension of traditional distillation method. For a fixed task, circuit distillation modifies the student's learning objective to both mimic the behavior of the teacher and the mechanisms used by the teacher to solve the task. A new metric is proposed to measure the distance between mechanisms across two models. A host of distillation experiments are performed using the Llama family of models (with sizes ranging from 1B to 8B) on two tasks: entity tracking and theory of mind. The paper finds that circuit distillation outperforms naive behavior distillation.

**Strengths:**

Overall, I found the paper easy to follow; the presentation is clear, and the treatment is relatively concise. Given existing literature on "mechanism" transfer across different tasks [Merullo et al. 2024](https://arxiv.org/abs/2310.08744), the motivation of the problem to me is very clear. The ablations explored make a convincing argument that the proposed circuit distillation is effective at transferring capabilities from the teacher to the student. Concretely,
- [Lines 40-45]: The motivation for the method is clearly and convincingly stated.
- [Table 1, 2]: Thorough ablations with the addition of the similarity metric are performed which demonstrate that the addition of circuit distillation unlocks more performance and alignment between the student and teacher.
- [Sections 4.1, 4.2]: Two different tasks that vary in both complexity and skill set are explored (entity tracking and theory of mind). This demonstrates that the method could be scaled to tasks that are even more complex.

**Weaknesses:**

## Major Issues

There are several issues with the paper in its current form.

My main qualm with the paper is its novelty. Two main contributions the authors claim are (1) "circuit *distillation*...modifies objective...to direct alignment of internal circuits" [Line 53]; and, (2) "a functional circuit mapping strategy" [Lines 70-72]. Both of these ideas have been extensively discussed in the mechanistic interpretability community and I believe there may be a lapse in the authors' literature review. More specifically,
- *Causal Abstraction* is essentially a mapping between fine-granularity components in one model (in the author's case, the teacher model) and coarse-granularity components (student model) that preserves the underlying algorithm. [Geiger et al. 2021](https://proceedings.neurips.cc/paper_files/paper/2021/file/4f5c422f4d49a5a807eda27434231040-Paper.pdf); [Geiger et al. 2025](https://www.jmlr.org/papers/volume26/23-0058/23-0058.pdf) formulate this mathematically and works such as [Geiger et al. 2023](https://arxiv.org/abs/2303.02536), [Gupta et al. 2024](https://arxiv.org/abs/2407.14494), [Sun et al. 2025](https://arxiv.org/abs/2503.10894) devise practical algorithms to learn these mappings.
- The mapping $\mathcal{M}$ that the authors propose in [Lines 187-199] is simply a special case of causal abstraction. Also, that this mapping needs to obey transformation invariances specified in [Lines 103-107] (orthogonal transformation, isotropic scaling) has also been considered by [Geiger et al. 2023](https://arxiv.org/abs/2303.02536) (see Definition 3).
- *Distilling circuits from one model to another.* The work of [Gupta et al. 2024](https://openreview.net/pdf?id=R9gR9MPuD5), I believe is another important omission. The authors take a model with a known circuit and train another model (of potentially different architecture) from scratch such that the circuits across both models are equivalent. Although the setup is slightly different, the ideas of causal abstraction are used to locate and enforce an adaptive alignment map. See [Gupta et al. 2024](https://openreview.net/pdf?id=R9gR9MPuD5) (see Algorithm 1), the authors also break the loss into two distinct parts: behavioral alignment which they call $L_{\mathrm{behavior}}$ , and circuit alignment which they call $L_{\mathrm{SIIT}}$ and $L_{\mathrm{IIT}}$.


The setup assumes that the student model is "warm" in that it can perform the task reasonably well before distillation. This assumption is not reasonable because often we are trying to distill capabilities from the teacher to the student that the student may not have learned before.
- The circuit on the student model needs to be identified [Lines 196-197]. But, a non-vacuous, unique circuit on the student model only exists if the student has some non-trivial performance on the target task; otherwise $\Delta P_s(h_s)$ could be constant for all $h_s$ and $d_{\mathrm{abl}}$ would be not informative ([Heimersheim and Nanda 2024](https://arxiv.org/pdf/2404.15255)).
- Further, the ablation studies the authors perform support this claim, as they demonstrate that it is necessary for the student circuit to be identified and paired properly with the teacher circuit for circuit distillation to achieve high performance (Table 1, 2, rows CE + Rand CKA).
## Minor Issues/Nitpicks

- The problem setup assumes that the teacher model must be whitebox since we need to identify its circuits and grab its internal representations during training [Lines 235-240]. This is much stronger than only requiring logits from the teacher model. Thus, it is unclear to me that knowledge distillation is a fair baseline comparison.
- Why define Equation 3, 4? They are never reference again in the main text and Equation 5 can be defined equivalently as $$d_{\mathrm{abl}}(h_s , h_t) = | P_{s\_\mathrm{abl}}(h_s) - P_{t\_\mathrm{abl}}(h_t) |.$$
- The authors claim that "our findings indicate that it is feasible to distill not just knowledge, but also the algorithms that produce it" [Lines 79-80]. I believe that this is an over-claim. The method only aligns individual components (where "align" is defined as aligning their representations). It's unclear
	- The circuits are even the same (i.e. ablating $h_s$ in the student model and $\mathcal{M}(h_s)$ in the teacher model result in the same drop in performance **after** circuit distillation).
	- Even if the circuits were the same, it seems quite a leap to go from individual circuit component equivalences to algorithmic equivalence (see Section 2.4 in [Heimersheim and Nanda 2024](https://arxiv.org/pdf/2404.15255)).

**Questions:**

* How would circuit distillation generalize to much more complex tasks like next-token prediction? In such a case, averaged over all next tokens the teacher circuit could be the entire model. How would $M$ nontrivially change to account for this?
* To clarify, on [Lines 233-234] when the authors say that "$y$ denoting the teacher labels," they mean that $y$ is the logits from the teacher model?

---

> ### Author Response · Authors · 2025-11-13
> **Response to Reviewer #1**
>
> We thank the reviewer for their time and for the detailed review. We are glad they find our motivation strong and ablations convincing. We believe some of the major concerns stem from a misunderstanding of our core contributions relative to the cited literature. While the reviewer cites relevant efforts (and we'll add discussion of them to the manuscript), **our primary contribution is using teacher internals to improve model distillation; these prior works do not concern model distillation.** We maintain that circuit *distillation* is therefore novel. We clarify below.
>
> **On Novelty (Major Issue #1)**
>
> We thank the reviewer for highlighting the valuable work on Causal Abstraction (Geiger) and related methods. However, our effort here is specifically about using whitebox interpretability to improve *distillation*; as far as we are aware, no prior effort (including the cited work) has proposed or evaluated methods that attempt to such transfer in context of model distillation.
>
> The Causal Abstraction framework of Geiger is a powerful *formalism* for verifying algorithmic equivalence, but these efforts do not discuss distillation at all. The question of *mechanism alignment* (a necessary step for our work) is related and we will integrate discussion accordingly. But again, these efforts are not concerned with distillation. The specific mapping strategy we propose here is incidental to our main contribution.
>
> The Gupta paper is also interesting and we agree *related*, but again does not specifically discuss *distillation*. Instead, this work offers a benchmark of which can be used to evaluate circuit discovery methods.
>
> Concretely, our contributions in this direction include:
> - Targeted, efficient transfer: Efficient capability transfer for distillation, not arguing for full algorithmic equivalence.
>
> - A distinct objective: We do not learn a general projection matrix. WE propose a new alignment loss for defined invariances, which is a different practical choice from Gupta et al. (which again, have a different aim in any case).
>
> In short, while Causal Abstraction is a valuable theoretical lens, our work provides a novel, practical *method for targeted, efficient distillation* by proposing a new mapping strategy and applying a CKA-based loss to a pre-identified circuit.
>
> **On "Warm Start" and Student Ckt ID (Maj Issues 2 & 3):**
>
> This appears to be a misunderstanding. We do not assume a special "warm" student. We use publicly available base models (e.g., Llama3-3B) as our students.
>
> The reviewer correctly notes (citing Heimersheim and Nanda) that a circuit can only be meaningfully identified if the model has non-trivial performance. This is not a weakness of our paper, but a *precondition that holds* in our setting. Our base student models *do* have non-trivial baseline performance (e.g., 0.68 and 0.72) from their pre-training. It is this existing, albeit weaker, capability that makes our method possible and effective.
>
> **On Minor Issues:**
>
> 1. **White-box Teacher**: The reviewer is correct. Our method is a white-box approach. Indeed, this is the central premise of our work! Our premise is that we can better distill teacher models by transferring their internal mechanisms to students. We believe that "standard distillation" (which looks at teacher outputs only) is indeed the appropriate baseline here because to our knowledge no prior work has established that whitebox methods can be helpful for distillation. If the reviewer of work that has established this which we could use for a different baseline, we would welcome the suggestion. .
>
> 2. **Eq 3 & 4**: In fact, we actually do reference Equations 3 and 4 again. Specifically, in Equation 5 we use the terms defined earlier.
>
> 3. **Over-claim (Algorithmic Equivalence)**: This is a fair point that we appreciate. Our claim to "distill... the algorithms" is imprecise. Our method enforces representational alignment at key circuit components, and our strong empirical results (especially the high performance of the isolated circuit post-distillation) support the hypothesis that this transfers the algorithm. We will revise our language to be more precise.
>
> **On Questions:**
>
> 1. **Generalization to Next-Token Prediction:** This is an interesting direction. Our work focuses on tasks with known, sparse circuits (Entity Tracking, ToM). For general next-token prediction, "the circuit" is likely the entire model, and our method would not apply in the same way. However, for more complex, bounded reasoning tasks that are solved via next-token prediction, one could apply our method if a sparse functional circuit were identified.
>
> 2. **Clarity on** $y$: We thank the reviewer for pointing this out. "$y$ is denoting the teacher labels. In our experiments, $L_{task}$ is a standard cross-entropy loss against the ground-truth dataset labels. We will correct this."
>
> We thank the reviewer again for their feedback and believe addressing these points will strengthen the paper.

---

> > ### Comment · Reviewer_Dpyn · 2025-11-25
> >
> > I acknowledge the authors' high-quality and timely response. The rebuttal has clarified a couple of misunderstandings that I have had. Namely, my concerns about the "warm start", "white-box" teacher have been addressed and I am also satisfied with the detailed answers to my questions.
> >
> > I thank the reviewers for revising the language to correct the potential over-claims. I have a few more points of suggestion:
> > 1. Regarding the novelty of the work, I acknowledge that both works I've cited previously in my review do not perform distillation. And, to the best of my knowledge, I do not know of any existing causal abstraction work that attempt to cross into this domain of distillation (a tangentially related work might be that of [Wu et al. 2023](https://proceedings.neurips.cc/paper_files/paper/2023/hash/f6a8b109d4d4fd64c75e94aaf85d9697-Abstract-Conference.html); [Wang et al. 2025](https://openreview.net/pdf?id=45EIiFd6Oa); but they analyze fine-tuning post-hoc).
> > 2. However, I do think there is a bit of mischaracterization of causal abstraction as a "formalism." Several works devise practical algorithms to grok this alignment ([Sun et al. 2025](https://openreview.net/pdf?id=6fDjUoEQvm); ). I would be more comfortable if the relation between these methods (specifically w.r.t. contribution "2" [Lines 70-72]) and your contributions are discussed in the manuscript (even if it is in the appendices).
> > 3. About the "complex-tasks" and how it may not generalize to general language modeling, I would be more comfortable if this were explicitly addressed in the main text (even if it were an aside).
> >
> > Honestly, I am not too familiar with the distillation literature and the assumptions that are commonly made in those problem settings. I do not think I can assess how practical this method will be to the average practitioner in that area. To this end, I may have been overconfident in my initial review and will defer to reviewers who may be more familiar with this body of work. I have adjusted my scores to reflect this and the authors' rebuttal. I appreciate the authors' patience during this discussion period and look forward to engaging further.

---

### Meta-Review · Area_Chair_TyzN · 2026-01-08

**Summary:**

Multiple reviewers Reviewers QV1G, Mq9k noted that the experiments were limited to narrow tasks (Entity Tracking, ToM) and relatively small models (Llama 1B-8B). There were concerns about whether this method could generalize.

Reviewers QV1G had issues on the readability of the introduction and Reviewers Mq9k had issues with following the paper and there was some misunderstanding of the paper by reviwer Dpyn.

Reviwers QV1G and 8ybk raised the issues of the circuit being known upfront and the student starting from a good starting point limiting the aplicablity of the method.

**Reviewer Concerns:**

Novelty: The rebuttal successfully clarified that while Causal Abstraction provides the formalism, this work is novel in applying white-box mechanistic interpretability specifically to the distillation objective (transferring capabilities to a student) rather than just verification or alignment of a single model. Reviewer Dpyn explicitly acknowledged this distinction in their post-rebuttal comment.

Methodological Assumptions: The authors clarified that "warm start" implies using standard open-weights base models (a standard practice) and that white-box access is the fundamental premise of mechanistic distillation.

Computational Cost: The authors provided concrete runtime data (~4 GPU-hours for the mapping step), addressing Reviewer Mq9k’s concern about the "heaviness" of the operation.

Heuristic Validity: The authors pointed to control experiments (Random CKA) to validatethe effectiveness of their mapping heuristic.

**Reviewer Scores:**

Dpyn 2 -> 6
QV1G 6 -> 6
Mq9k 4  -> 4
8ybk 4 - > 4


This places the paper in the borderline territory with presentation issue giving my hesitation for acceptance.

---

### Decision · Program_Chairs · 2026-01-26

Reject